This is just a preview and not the published paper.





# Stable isotopic evidence for high microbial nitrate throughput in a High Arctic glacial catchment

A. H. Ansari[1,2]

[1]Birbal Sahni Institute of Palaeobotany, 53 University Road, Lucknow - 226007, India
[2]Department of Geography, University of Sheffield, Sheffield, S10 2TN, UK

*Correspondence to*: A.H.Ansari (a.h.ansari@bsip.res.in)

**Abstract:** During summer, streams in the Arctic redistribute and export the solute derived from snowpack (accumulated by atmospheric deposition) melting and sediment weathering in its flowpath. The redistribution of dissolved nitrogen undergoes biogeochemical processing (nitrate production and consumption in flowpath). To assess the quantitative impact of these processing, $NO_3^-$-N, $NH_4^+$-N, total dissolved nitrogen (TDN) and stable isotope composition of the snowpacks was compared with the subglacial and proglacial stream waters. Snowpack derived dissolved organic nitrogen (DON) and $NH_4^+$-N provided the most probable substrate for additional $NO_3^-$-N produced by microbial nitrification. The flux of microbially produced and assimilated $NO_3^-$-N in the eastern and western proglacial streams were $1.64\pm1.41$ kg Day$^{-1}$ and $1.41\pm1.43$ kg Day$^{-1}$ and, $1.39\pm1.41$ kg Day$^{-1}$ and $1.35\pm1.43$ kg Day$^{-1}$ respectively. These overwhelming amounts of $NO_3^-$-N production and assimilation reveals a hitherto unknown level of microbial processing in the Arctic glacial ecosystem. The balance between the two microbial processes and consistently low dissolved inorganic nitrogen ($NH_4^+$-N + $NO_3^-$-N) in the proglacial streams indicate a fast in-stream recycling of assimilated $NO_3^-$-N however, the fate of such $NO_3^-$-N remains unresolved.

Keywords: Subglacial, Proglacial, Stable isotope composition, Nitrification, Assimilation

## 1. Introduction

Meteorological and atmospheric studies from Svalbard (a Norwegian archipelago in the Arctic Ocean) have established that atmospheric circulation transports substantial amount of gaseous and particulate pollutants to this relatively pristine Arctic environment (Barrie, 1986; Law and Stohl, 2007; Stohl, 2006). Among the transported pollutants, nitrate has been a major focus for polar researchers due to its well known role in ozone chemistry (Crutzen et al., 1992; Solomon et al., 1996; Waibel et al., 1999) and acid rain events (KERR, 1979; SCHINDLER, 1988; Wright et al., 1988). However, the role of transported nitrate in the Arctic glacial ecosystem with changing glacial coverage remains understudied. This is probably because the outstanding proportion of atmospheric nitrate deposition, by and large happens amid the winter-spring when the greater part of the Arctic is snow covered (Eneroth et al., 2003; Hodson et al., 2009a), and as soon as melting starts in the summer, nitrate laden snowpacks disappear quickly in the form of streams that discharge into the sea. Nevertheless, the higher altitude areas are commonly covered by perennial glaciers that generally receive greater precipitation. During summer, melting of these higher altitude snowpacks maintain water discharge in major proglacial channels for longer periods (2-3 months) (Ansari et al., 2012; Hodson et al., 2005a; Rutter et al., 2011) allowing a greater interaction between the deposited nitrate and geobiological component of the flowpath. Since it has been generally considered that due to low temperature biotic impacts on nitrogen cycling in these streams have low quantitative significance, very little consideration have been paid towards the quantitative role of major biological processes, i.e. nitrification, denitrification, and nitrate assimilation, etc. However, recent developement in understanding of life in glacial ecosystem (Anesio and Laybourn-Parry, 2012; Hodson et al., 2008; Stibal et al., 2012) has demonstrated higher biological activities than previously thought. In the light of theses findings, now glaciers (that covers ≈ 10 % land mass of our planet) are perceived as an ecosystem with implications for global nutrient cycling (Hood et al., 2015). But, quantitative estimate for biological nitrogen turnover in a glacial ecosystem is still lacking. Such estimates will help to understand the magnitude of biological response to atmospheric nitrogen deposition with changing glacial coverage, especially as the majority of glaciers are currently encountering protracted wet periods that provide water for microbial activity (Anesio et al., 2009).

Mass balance studies of discharge weighted nitrogen for the glacial streams, and temporal nitrogen dynamics in soil or static water bodies have been used as a classical tool to infer possible biogeochemical processes (Brooks

This is just a preview and not the published paper.


et al., 1998; Hodson et al., 2005b; Petrone et al., 2007; Tranter, 1991) i.e. nitrate gain as nitrification and loss as denitrification or assimilation. The outcome of these studies has also been supported by molecular investigation of samples from supraglacial, subglacial and proglacial regions that has demonstrated the presence of microbiota like, ammonium oxidising archaea, nitrifier, denitrifiers etc. (Boyd et al., 2011; Jakub et al., 2013). The magnitude of different microbial processes varies with seasonal changes in the local environment(Hodson et al., 2009b). For example, low level of dissolved oxygen can often be seen in the early subglacial upwelling water that demonstrates poor ventilation between atmosphere and old subglacial water stored since last summer (Hodson et al., 2005a). Such old subglacial waters are generally nitrate poor with positive $\delta^{15}$N-NO$_3$ values; a signature for denitrification (Ansari et al., 2012; Wynn et al., 2006; Wynn et al., 2007). A comprehensive modelling study by Roberts et al. (2010) suggests that microbial assimilation is a common process for nitrate removal from subglacial waters, whereas, denitrification accounts for only a small fraction of nitrate removal. Surface waters (supraglacial and proglacial streams) in the glacial ecosystem have been frequently reported for nitrate production by microbial nitrification of snowpack derived ammonia and dissolved organic nitrogen (Ansari et al., 2012; Hodson et al., 2005a; Hodson et al., 2009b; Hodson et al., 2005b).

Microbial nitrogen fixation has only been detected from supraglacial cryoconite holes (Telling et al., 2011). To determine the magnitude of these microbially mediated processing of the deposited nitrogen, quantitative estimation of microbial nitrate production and consumption is required. Until now no such estimation has been produced for the glacial ecosystem. So, this study measured the nitrate concentration, discharge, and stable isotopic composition of nitrate in the two noteworthy glacial streams to ascertain the hitherto unknown flux of microbially mediated nitrate production and consumption.

Stable isotope tracers are a very useful technique to distinguish various sources of nitrogen in aquatic ecosystem studies (Kendall et al., 2007). The isotopic identification of nitrogen source and associated processes depends upon the two parameters: 1) isotopic signature of source: different source of the nitrogen often has a unique isotopic composition that makes the basis of their identification. 2) Isotopic fractionation: many of the processes often changes the isotopic ratio in the anticipated direction, therefore can be easily traced from the final isotopic composition. The $^{15}$N approach has been instrumental since its early use in biogeochemical studies (Heaton, 1986), later since last one decade use of $^{18}$O along with $^{15}$N has improved the employability of this technique in atmospheric studies (Amberger and Schmidt, 1987; Curtis et al., 2011; Wankel et al., 2006; Wynn et al., 2007). The use of this dual isotope technique in Midtre Lovenbreen glacier basin helped to identify the nitrate sources and the biological process and provided the preliminary model of glacial nitrogen cycling (Ansari et al., 2012).

The goals of this study are: 1) to understand the influence of the in-stream biogeochemical processes on dissolved nitrogen pool by evaluating downstream changes in DON, NH$_4^+$-N, NO$_3^-$-N concentration and stable isotope composition of nitrate. 2) Quantitative estimation of NO$_3^-$-N produced and consumed by nitrification and assimilation/denitrification respectively by standard mass-balance calculation.

## 2. Field Site and Methodology

### 2.2. Midtre Lovénbreen

Midtre Lovénbreen (78.53° N and 12.04° E) is a 6 km long glacier that occupies an area of approximately 5.5 km², on the Brøggerhalvøya peninsula in North West Spitsbergen (Figure 1). Its height ranges from 50 to 600 m above sea level and maximum depth of this glacier is approximately 180 m. Due to high ice overburden pressures, a large portion of the glacier bed is at the pressure melting point and therefore maintains subglacial drainage all year. During winter, some subglacial meltwater gets trapped until a subglacial upwelling forms near the glacier snout (usually in July). As this subglacial system is well connected with the higher altitude crevasses, subglacial upwelling increasingly discharges fresh meltwater supplied by snowpack melting above the crevasses.

Almost every summer (from middle or late May) glacial ablation results in three major proglacial streams in the Midtre Lovénbreen catchment. One stream each emerges from the eastern lateral (MLE) and western lateral side (MLW) of the glacial snout, respectively, and a third one emerges from the middle of the glacial snout. The middle stream disappears earlier than the MLE and MLW. These streams receive discharge from supraglacial snow-melting and marginal ice melting and meander over short lateral distances, but follow the same incised flow path almost every year. As a result deep incisions of streams are typical over the proglacial field. All of the three streams discharge into Kongsfjorden, following a short interaction with the proglacial forefield (Figure 1).

This is just a preview and not the published paper.





*2.3. Sampling site and sampling*

The eastern proglacial stream (MLE) received water from two sources: i) ice marginal stream (MLE3), mostly
fed by glacial ice melt and ii) subglacial runoff opening at MLU [sometime before DOY 186 (5th July 2010)].
The waters from MLU and MLE3 mixed downstream and constituted the resultant discharge and chemistry for
MLE2. From MLE2 to MLE1 the stream received no other visible discharge. However, the source of water in
the western stream (MLW) was largely ice and snow melt, which emerged from the glacier margin and travelled

downstream via MLW3, MLW2 and MLW1. MLW received no other visible discharge between MLW3 and
MLW1. Therefore, this study believes that in the MLE discharge from MLE2 to MLE1, and in the MLW
discharge from MLW3 to MLW1 remained unchanged and changes in solute composition of the streams
between these sites were caused by factors working within the main channel, most probably weathering and
microbial processes.

Five bulk snowpack samples were collected from the center line of the glacier surface at 80 to 90 m elevation
intervals (SG1 to SG5 in Figure 1) on DOY 160 (9th July 2010). Thereafter, sampling was designed to
investigate the downstream biogeochemical changes in the streams using 3 sites (MLW1, MLW2, MLW3) on
the western stream and 4 sites (MLU, MLE1, MLE2, MLE3) on the eastern stream (Figure 1) on DOY 186. Five
weekly profile samples were collected from the above stream sites throughout the summer until DOY 227 (15th

August 2010). Snow (thawed at 4° C before filtration) and stream samples were filtered using 0.45 µm pore size
cellulose nitrate filter papers at the British NERC Arctic station lab facility (1-2 hour after sample collection).
Filtered samples were stored in 60 mL polyethylene bottles refrozen at -20° C and transported back to the
laboratory for chemical and isotopic analysis. Bottles were pre-rinsed with filtered sample water before sample
transfer.


*2.4. Hydrological measurements*

Water pressure and conductivity were logged at hourly intervals at MLE1, MLW1 and MLU by using In Situ
Troll 400 and 9500 sensors. However, these systems were ill-equipped to deal with the dynamic sediment-rich
streams, making it impossible to present data from MLU. Meltwater discharge at MLW1 and MLE1 was

established after calibrating the pressure transducer records using the salt dilution method. This involved the
injection 300 to 500 g of NaCl following the methodology reported by Day (1977). According to the standard
error of regression models fitted to the water pressure and discharge data errors were 15 %.

*2.5. Chemical analysis*

Major ions (anions: $NO_3^-$-N, $Cl^-$, $SO_4^{2-}$; cations: $NH_4^+$-N, $Na^+$, $K^+$, $Ca^{2+}$, $Mg^{2+}$) were measured using a Dionex
DX 90 ion chromatograph, operated through a 4400 integrator and AS40 autosampler. Repeatability for mid-
range standards (calibration range: 0 to 2 $mgL^{-1}$) was 1.6, 1.4, 5.7 % for the anions listed above and 2.5, 0.06,

0.2, 0.08, 1.5 % for the cations respectively. Precision errors deduced from repeat analyses of separate filtered
aliquots from a single snow sample were less than 5 %. A detection limit of 1 $µgL^{-1}$ was imposed upon all
chromatograms, and all analytical blank results were consistently below this limit. Total dissolved nitrogen
(TDN) was analyzed following the persulphate oxidation method (Solórzano and Sharp, 1980) with a precision
error of ±10 % or less, according to mid-range standards (calibration range: 0.1 to 1 $mgL^{-1}$). Detection was

achieved using a Skalar SAN++ continuous flow analyser which employ a cadmium column for nitrate
reduction to nitrite. Dissolved organic nitrogen (DON) was then calculated by subtracting inorganic nitrogen
($NO_3^-$-N + $NH_4^+$-N) from TDN.

*2.5.1. Denitrifier method for $^{15}N$ and $^{18}O$ isotope analysis*

This method is based on denitrification, developed by Sigman et al. (2001) and Casciotti et al. (2002) and
amended by Kaiser et al. (2007). It is rapid, can be fully automated and requires only ≈ 10 nmole $NO_3^-$, 2-3
orders of magnitude less than the silver nitrate method, at a similar analytical precision (0.2 ‰ for $\delta^{15}N$, 0.5 ‰

for $\delta^{18}O$). It is based on denitrification of the $NO_3^-$ to nitrous oxide ($N_2O$) by a specific denitrifier *Pseudomonas
aureofaciens*. This strain lacks active nitrous oxide reductase required for converting nitrous oxide to free
nitrogen and shows little isotope exchange between water and intermediate species during denitrification. The
$^{15}N/^{14}N$ and $^{18}O/^{16}O$ ratios of the $N_2O$ were determined by mass spectrometry (Sercon TG II/Geo) and corrected
to $\delta^{15}N$ and $\delta^{18}O$ values relative to atmospheric-$N_2$ and VSMOW (Viena Snow Mean Ocean Water) respectively

through comparison with the nitrate reference materials IAEA-$NO_3$, USGS-34 and USGS-35.

This is just a preview and not the published paper.





Therefore, this method is largely dependent upon microbial preprocessing before analysis. Bacterial preparation includes the following steps:


*2.5.2. Media preparation*

For media preparation 1.8 g $KNO_3$, 0.45 g $(NH_4)_2SO_4$, 11.7 g $K_2HPO_4$, and 54 g Triptych Soy broth granules were properly dissolved at room temperature in 1800 mL of MilliQ water (18 Ω) within a 2 L Duran or glass beaker. Prepared media were then distributed to 4 x 500 mL bottles in equal amount (≈ 445 mL each). These bottles with the media and stopper were separately covered with aluminium foil and autoclaved for 1.5 h thereafter left within an autoclave to cool. Immediately after removing from autoclave media bottles were capped and sealed with crimp. These media bottles were kept under dark at room temperature.


*2.5.3. Plate preparation*

30 g of Tryptic soy agar granules were added to 500 mL of freshly prepared media (not necessary to be incubated) and heated along with continuous stirring to dissolve agar properly. Dissolved agar solution autoclaved in Duran bottle for 0.5 h. This solution was removed from the autoclave before it cooled down 40° C (before it freezes) and poured into plate to fill it ¾ and swirl round to spread the agar solution on plate base equally in a laminar flow hood. Plates with agar solution were left semi-opened overnight to condense in laminar. These plates were closed and then sealed with parafilm to store in fridge. Plates with any biological growth in it were disposed.


*2.5.4. Preparation of nutrient free media (NFM)*

To prepare NFM, 0.5 g $(NH_4)_2SO_4$, 13 g $K_2HPO_4$, and 60 g Triptych Soy broth granules were dissolved properly at room temperature in 2 L MilliQ water (18 Ω) in a glass or duran beaker. Prepared solution was transferred and into 250 mL duran bottles and autoclaved with loosen cap for 0.5 h. Thereafter, these bottles were removed from the autoclave and their caps were tightened to store at room temperature. NFM with cloudy appearance were disposed.


*2.5.5 Culturing bacteria on plates*

A dollop of the frozen bacteria from 2.5 mL eppendorfs was transferred onto inoculating point at prepared plates using a sterilised 100 μL pipette tip or toothpick. Immediately after transfer appendix was closed and put back into freezer. Following this, bacteria were streaked over plate starting from inoculating point. Then the plate was closed and sealed with parafilm to incubate in dark and room temperature (or controlled temperature 15° C) for 3-4 days.

*2.5.6. Bacterial inoculation*

Bacterial inoculation was done about one week before their use for sample denitrification. To maximize the chance of growth, bacteria were inoculated in pairs; two 9 mL media tubes from a single colony of freshly grown plates (colony was transferred with sterilised loop). Inoculated media tubes were capped and sealed with paraffin, then incubated overnight at room temperature on the shaker in horizontal position. Two incubated tubes were mixed in a bigger tube that was used to inoculate the media bottles. Center of crimp seal of media bottle lifted and 2.7 mL of freshly incubated culture from the bigger tubes were injected into the media bottle. These bottles were kept on shaker for 6-10 days.


*2.5.7. Concentrating the bacteria*

Culture from the bottle (500 mL) was divided into two 250 pre-autoclaved centrifuge bottles and centrifuged for 10 minutes at 4950 rpm. Pink-white colored bacterial aggregate settled at the bottom of centrifuge bottles. Supernatant liquid was removed from the bottle, leaving bacterial aggregate and bottles kept upside down to

This is just a preview and not the published paper.





separate any further liquid left. Then bacterial aggregates from 2 bottles were resuspended in about 225 mL of NFM media and shook. 45-70 drops of antifoam agent were added to avoid frothing before during shaking.

*2.5.8. Sample vial preparation and purging*

3 mL of concentrated bacterial solution was pippeted to 20 mL vials and then vials are closed with butyl rubber stopper and crimped. These vials were purged for 45 minutes with He gas at a flow rate of 60-70 mL min⁻¹. After this first purge vials were kept on shaker on sideways for 5 h or overnight and then again purged it for 45
minutes with He gas.

      *2.5.9. Sample injection*

Glass tight syringe was rinsed with MilliQ water (18 $\Omega$) for 10 times and 3 times with sample before each sample injection. Sample (amount 10 nanomoles of $NO_3^-$) was injected through butyl stopper. At the same time if the sample volume exceeds 1 mL a vent needle (25 gauge, leak tight) was used to remove the excess pressure developed within the vial (sample volume should not exceed 5 times of cell concentrate). Subsequently vials were incubated at room temperature for 6 h in an inverted position to check any leakage. At the end of this
incubation, 0.1-0.2 ml of 6 M NaOH was injected to each vial and shook to lyse the bacteria.

      *2.5.10. Analysis at IRMS*

Generated nitrous oxide gas (as a result of bacterial denitrification) within the vials were extracted by an automated extraction system; purging with carrier gas He (flow rate 15-20 mL/min). Purged gas was then passed with nafion drier column (to remove water molecules) and ascarite drier column (to remove $CO_2$). Ahead in the line, purified gas trapped in a U shaped tube filled with glass beads and immersed in liquid nitrogen. This trapped $N_2O$ released when trap comes out of liquid nitrogen and its temperature increased. From here passing
through another nafion drier $N_2O$ gas was separated in the GC column within GC then finally swept to IRMS for isotope analysis.

*2.6. $\delta^{18}O$-$H_2O$ analysis*

Water $^{18}O/^{16}O$ ratios were determined on $CO_2$ equilibrated with water samples in an Isoprep 18 coupled to a SIRA mass spectrometer (Micromass, Middlewich, England). The ratios are reported as $\delta^{18}O$ values versus VSMOW, based on comparison with laboratory standards calibrated against VSMOW and SLAP, with analytical precision (1 standard deviation) typically better than 0.05 ‰.

## 3. Results

      *3.1. Hydrological conditions*

During summer 2010, discharge data were collected from DOY 184 (3rd July 2010) to 238 (26th August 2010) at MLE1 and MLW1. The proglacial eastern stream received discharges from two distinct sources of melt water: the eastern ice marginal stream (MLE3) and subglacial runoff (MLU). The MLW1 had no subglacial runoff, and so supraglacial snow and marginal ice melt remained the predominant source of water throughout the summer.
The water discharge at MLE1 initially reached a maximum of 1.5 m³s⁻¹ during early summer [on the DOY 194 (13th July 2010)] and afterwards decreased to 0.13 m³s⁻¹ during late summer (on the DOY 238). Similarly, the water discharge at MLW1 also decreased from the early to late summer (0.3 m³s⁻¹ to 0.1 m³s⁻¹) but the discharge and its variation was lower than at MLE1 (Figure 2).

      *3.2. Snow and stream nitrogen composition*

In the snowpacks samples $NO_3^-$-N, $NH_4^+$-N and DON ranged between 12.8-29.3 µg L⁻¹, 17.8-26.1 µg L⁻¹ and 73.6-93.6 µg L⁻¹ respectively. The MLE and MLW those were discharged by snowpack melting had slightly
different concentrations; at the MLE sites $NO_3^-$-N ≈ 1.0-52.4 µg L⁻¹, $NH_4^+$-N ≈ b.d.-11.6 µg L⁻¹ and DON ≈

This is just a preview and not the published paper.





30.2-108.8 µg L⁻¹ (Figure 3); at the MLW sites $NO_3^-$-N ≈ 0.3-31.2 µg L⁻¹, $NH_4^+$-N ≈ b.d.-7.5 µg L⁻¹ and DON ≈ 32.3-103.6 µg L⁻¹ (Figure 4). However, estimated snowpack-derived nitrogen concentration for the streams: $NO_3^-$-N ≈ 5.7-53.4 µgL⁻¹, $NH_4^+$-N ≈ 5.5-51.6 µgL⁻¹ were substantially different from their actual concentration (Table 1).


The equation used for the calculation of snowpack derived nitrogen (SN) is:

$$SN = \left[ \frac{\sum_{i=1}^{n} (N_{Snowpack}/Cl^-_{Snowpack})_i}{n} \right] \times Cl^-_{Stream} \qquad (1)$$

Where, $N_{Snowpack}$, $Cl^-_{Snowpack}$ and $Cl^-_{Stream}$ denote concentration of inorganic nitrogen ($NO_3^-$-N, $NH_4^+$-N), chloride concentration in the snowpacks and chloride concentration in the streams respectively.

### 3.3. $\delta^{15}N(NO_3)$, $\delta^{18}O(NO_3)$ and $\delta^{18}O(H_2O)$ composition of snowpacks and streams


In the snowpacks $\delta^{15}N(NO_3)$ and $\delta^{18}O(NO_3)$ varied from -13 to -15 ‰ and 87 to 90 ‰ respectively. During the following summer $\delta^{15}N(NO_3)$ and $\delta^{18}O(NO_3)$ of the streams varied from -9.6 to 6 ‰ at the MLU and from 7 to 73.5 ‰ at other stream sites (Figure 2 and Table 1). On spatial scale, $\delta^{15}N(NO_3)$ either remained constant or showed a change of up to 4.7 ‰ over a course of ca. 1 km distance between the ice marginal and downstream

sites, whilst changes in $\delta^{18}O(NO_3)$ were regular that reached up to -16 ‰. The $\delta^{15}N(NO_3)$ and $\delta^{18}O(NO_3)$ from all the samples were significantly correlated in which positive enrichment of $\delta^{15}N(NO_3)$ and contemporary negative enrichment in $\delta^{18}O(NO_3)$ formed an apparent linear relationship [excluding positive $\delta^{15}N(NO_3)$ value, Pearson correlation coefficient for the rest was -0.95 (p = 0.0001, n = 34)] (Figure 5).
Figure 3a-e  and 4a-e demonstrates spatial variations in $NO_3^-$-N, DON+$NH_4^+$-N, $\delta^{15}N(NO_3)$, and $\delta^{18}O(NO_3)$.

$NO_3^-$-N concentrations between MLE2 and MLE1 remained unchanged on DOY 186-227. However, positive change in the $\delta^{15}N(NO_3)$ and negative change $\delta^{18}O(NO_3)$ occurred on DOY 207 (26th July 2010)-227 and on DOY186, 207, and 218 (6th August 2010) respectively. DON+$NH_4^+$-N concentration between MLE2 and MLE1 increased on DOY 186-227. $NO_3^-$-N concentrations between MLW3 and MLW1 remained unchanged on DOY 186-227 but,  positive change on $\delta^{15}N(NO_3)$ and negative change in $\delta^{18}O(NO_3)$ occurred on DOY 186-

227. DON+$NH_4^+$-N concentration between MLW3 and MLW1 decreased on DOY 186, 227and increased on DOY 198 (17th July 2010)-218.

### 3.4. Calculation of microbially produced and consumed nitrate in the flowpath


This study used equation derived on the mass balance principle to calculate the $NO_3^-$-N content: i) produced by microbial process and simultaneously, ii) assimilated/denitrified by the same or different microbes in the flowpath. As we know from the earlier studies (Ansari et al., 2012)3) that this catchment has two major sources of nitrate: a) snowpack melting; b) microbial production. The aggregate $\delta^{18}O(NO_3)$ and $\delta^{15}N(NO_3)$ of total

nitrate (snowpack-derived + microbial-derived) can be calculated by mass balance equation (3) and (4) respectively (in the equations *sn* denotes snowpack-derived, *mi* denotes microbial-derived, *st* denotes stream-derived, *asm/den* denotes assimilated/denitrified and *C* denotes concentration). Then changes in aggregate $\delta^{18}O(NO_3)$ and $\delta^{15}N(NO_3)$ due to microbial consumption in the flowpath can be calculated by subtracting $\delta^{18}O(NO_3)$ and $\delta^{15}N(NO_3)$ of streams from the aggregate value (see equation 5 and 6). Once the microbial

nitrate is known, total nitrate budget for the stream can be calculated by following equation.

$$C_{st} = C_{sn} + C_{mi} - C_{asm/den} \qquad (2)$$

$$\delta^{18}O(NO_3)_{(sn+mi)} = \frac{\delta^{18}O(NO_3)_{sn} \times C_{sn} + \delta^{18}O(NO_3)_{mi} \times C_{mi}}{(C_{sn} + C_{mi})} \qquad (3)$$

$$\delta^{15}N(NO_3)_{(sn+mi)} = \frac{\delta^{15}N(NO_3)_{sn} \times C_{sn} + \delta^{15}N(NO_3)_{mi} \times C_{mi}}{(C_{sn} + C_{mi})} \qquad (4)$$


This is just a preview and not the published paper.





$$\Delta^{18}O(NO_3) = \frac{\delta^{18}O(NO^3)_{sn} \times C_{sn} + \delta^{18}O(NO^3)_{mi} \times C_{mi}}{(C_{sn} + C_{mi})} - \delta^{18}O(NO^3)_{st} \qquad (5)$$

$$\Delta^{15}N(NO_3) = \frac{\delta^{15}N(NO^3)_{sn} \times C_{sn} + \delta^{15}N(NO^3)_{mi} \times C_{mi}}{(C_{sn} + C_{mi})} - \delta^{15}N(NO^3)_{st} \qquad (6)$$

Assuming microbial assimilation/denitrification was the major $NO_3^-$-N consuming process in this catchment as suggested by Roberts et al., (2010), Wynn et al., (2006, 2007) and (Ansari et al., 2012), and because microbial assimilation/denitrification changes the $\delta^{18}O(NO_3)$ and $\delta^{15}N(NO_3)$ mostly in 1:1 ratio (Kendal et al., 2007), the equation (5) /equation (6) will derive equation (8).

$$\frac{\Delta^{18}O(NO_3)}{\Delta^{15}N(NO_3)} = \frac{1}{1} = \frac{\delta^{18}O(NO_3)_{sn} \times C_{sn} + \delta^{18}O(NO_3)_{mi} \times C_{mi} - \delta^{18}O(NO_3)_{st} \times (C_{sn} + C_{mi})}{\delta^{15}N(NO_3)_{sn} \times C_{sn} + \delta^{15}N(NO_3)_{mi} \times C_{mi} - \delta^{15}N(NO_3)_{st} \times (C_{sn} + C_{mi})} \qquad (7)$$

$$C_{mi} = \frac{\delta^{18}O(NO_3)_{sn} \times C_{sn} - \delta^{15}N(NO_3)_{sn} \times C_{sn} - \delta^{18}O(NO_3)_{st} \times C_{sn} + \delta^{15}N(NO_3)_{st} \times C_{sn}}{\delta^{18}O(NO_3)_{st} + \delta^{18}O(NO_3)_{mi} - \delta^{15}N.(NO_3)_{st} - \delta^{18}O.(NO_3)_{mi}} \qquad (8)$$

Total $NO_3^-$-N flux of the day ($F_{dtot}$ in $g\ Day^{-1}$) was then estimated from the concentration of the single daily sample ($C_i$ in $\mu g\ L^{-1}$) and mean of daily discharge ($Q$ in $m^3\ sec^{-1}$) by using the following equation.

$$F_{dtot} = \sum_{i=1}^{n}(Q_{dtot} \times C_i) \qquad (9)$$

Since for these streams diurnal variations in $Ci$ were found insignificant (Hodson et al., 2005b), one sample for the day in this study considered adequate to elucidate the daily nitrogen flux. The analytical errors of each parameter and standard deviation of calculating mean values were integrated into the above mentioned calculations using the standard error propagation method (detailed formulae are given in the appendix 1 A, B, & C).

In this way microbial production of and assimilation of $NO_3^-$-N was calculated for most of the stream samples. Figure 6 shows snowpack-derived $NO_3^-$-N at MLE1 decreased from 5.35±0.80 kg $Day^{-1}$ on DOY 186 to 0.54±0.08 kg $Day^{-1}$ on DOY 227. Microbially derived $NO_3^-$-N flux at MLE1 first decreased from 2.02±1.34 kg $Day^{-1}$ on DOY 186 to 1.50±1.36 kg $Day^{-1}$ on DOY 198 after that flux varied between 1.36±1.33 and 1.75±1.03 kg $Day^{-1}$. Microbially consumed $NO_3^-$-N flux ranged between 1.02±1.32 kg $Day^{-1}$ on DOY 186 to 1.97±2.0 kg $Day^{-1}$ on DOY 218. Snowpack-derived $NO_3^-$-N flux at MLW1 first increased from 1.62±0.24 kg $Day^{-1}$ on DOY 186 to 1.84±0.28 kg $Day^{-1}$ on DOY 198. Thereafter the flux decreased to 1.07±0.16 kg $Day^{-1}$ on DOY 207 to 0.27±0.04 kg $Day^{-1}$ on DOY 227.

## 4. Discussion

### 4.1. Stable isotope composition of probable nitrogen substrates

The $\delta^{15}N$ value of $NH_4^+$-N in this region varies from -1.7 to -2.8 ‰ and $\delta^{15}N$ of rock 7.2 to 7.7 ‰ (Wynn et al., 2007). The $\delta^{15}N$ of local soil matter, organic ranges between 1.2 and 2.7 ‰ (Tye and Heaton, 2007) and of marine derived DON (from North Atlantic Ocean) ranges between 4 and 5 ‰ (Knapp et al., 2010). Thus, assuming fractionation during microbial conversion of these substrates (snowpacks derived $NH_4^+$-N, DON and PN) to nitrate in the Arctic environment is negligible (Tye and Heaton, 2007), $\delta^{15}N$ of $NO_3^-$-N produced this way is likely to range between -2.8 and 7.7 ‰. Mixing of this additional $NO_3^-$-N with snowpack derived $NO_3^-$-N ($\approx$ -14.1 ‰) defines the final $\delta^{15}N(NO_3)$ composition ranging between -14.4 and 5 ‰ (Figure 5). An increase in the fraction of additional $NO_3^-$-N thus increases in the accumulative $\delta^{15}N(NO_3)$ often observed at MLU and other proglacial stream sites. Although no exact value for the microbially produced nitrate is known from the catchment, this study replaced $\delta^{15}N(NO_3)_{mic}$ with the most likely end number, 0 ‰ considering the $\delta^{15}N(NO_3)$ of available nitrogen substrate for nitrate production (Figure 5).

This is just a preview and not the published paper.


### 4.2. Stable isotope composition of microbial-nitrate

The findings of Aleem et al. (1965), Hollocher et al. (1981), Andersson and Hooper (1983) and Kumar et al. (1983) suggested that nitrification process incorporates two $H_2O$-derived and one $O_2$-derived oxygen atom to form nitrate. Further, these studies reported no isotopic effect during oxygen incorporations or any further isotopic exchange. Since then this concept has been widely used to understand and quantify microbial nitrate in the aquatic and sedimentary environment (Anisfeld et al., 2007; Ansari et al., 2012; Kendall, 1998; Tye and Heaton, 2007; Wexler et al., 2011).


$$\delta^{18}O(NO_3)_{nitrification} = 2/3\ \delta^{18}O(H_2O) + 1/3\ \delta^{18}O(Atm.O_2) \tag{10}$$

The $\delta^{18}O(H_2O)$ (ranged from -9.5 to -11.7 ‰, mean = -10.5±0.5 ‰) and $\delta^{18}O$-Atm.$O_2$ [24 ‰, after Luz and Barkan (2011)] both are known. Thus $\delta^{18}O(NO_3)_{mi}$ for this study was calculated through equation (10). The
calculated $\delta^{18}O(NO_3)_{mi}$ ranged from 1.6 to 0.2 ‰ (mean 0.88±0.36 ‰). But, it must be noted that some recent studies have demonstrated microbial nitrate with $\delta^{18}O$ values both lower and higher than would be calculated from Equation (10) (Buchwald and Casciotti, 2010; Kool et al., 2011; Snider et al., 2010; Spoelstra et al., 2007). This has certainly overcast the confidence in the traditional use of equation (10) to understand the microbial-derived nitrate and therefore needs caution while interpreting the calculated value. The tight correlation between
the $\delta^{15}N(NO_3)$ and $\delta^{18}O(NO_3)$ (Figure 5) suggests that oxygen and nitrogen isotope participated in a constant ratio to form nitrate. Because change in nitrogen isotope composition of the nitrate after its formation is very unlikely, the oxygen isotope composition will also have to be consistent to maintain such a tight correlation. In this scenario the lowest value of $\delta^{18}O(NO_3)$ must be the closest value to the $\delta^{18}O(NO_3)_{mi}$ however, there is good chance that this lowest value represents microbial + snowpack derived nitrate. Following this discussion the
$\delta^{18}O(NO_3)_{mi}$ calculated through equation (10) seems plausible, therefore, applied in this study for quantitative estimation of microbial nitrate using equation (8).
The plot between $\delta^{15}N(NO_3)$ and $\delta^{18}O(NO_3)$ (Figure 5) demonstrates two end-members with unique isotopic values. The first end member is snowpack derived nitrate that can be easily distinguished due to its highly positive $\delta^{18}O$ (mean +89.1±1.3 ‰) and negative $\delta^{15}N$ (mean -14.1±0.82 ‰) unique to atmospheric nitrate. The
second end member is defined by $\delta^{18}O$ of microbially produced nitrate (mean 0.9±0.4 ‰). Most of the $\delta^{15}N(NO_3)$ and $\delta^{18}O(NO_3)$ of stream samples lies on or close to the adjoining line of the two end members represent that stream carried both snowpack-derived and microbial-derived nitrate simultaneously. Further, assuming no oxygen exchange between nitrate and water molecule, the isotopic data indicates that the proportion of microbial nitrate increased downstream as well as with the passing summer (Figure 3, 4, 5 and 6).
This gradual shifting of stable isotope values closer to the microbial signature could be either because of the microbial recycling of snow derived nitrate (nitrate used in microbial metabolism release as particulate or dissolved organic nitrogen and the mineralised back to nitrate) or mineralisation/nitrification of dissolved PN, DON and $NH_4^+$-N or both.


### 4.3. Proglacial and subglacial stream nitrogen cycling

The dissolved organic and inorganic nitrogen, isotopic composition along with other major ions were examined in the snowpacks and their subsequent summer meltwater in the subglacial and proglacial streams. Because $Cl^-$
is considered a conservative species the glacial watershed system (Tranter et al., 1996), the $NO_3^-$-N/$Cl^-$, $NH_4^+$-N/$Cl^-$, DON/$Cl^-$ ratios of snowpacks were used to quantify the snow-derived content of the solutes in the proglacial streams. That provided a general understanding of loss or gain of dissolved organic and inorganic nitrogen in the flow path. However, the actual value of gain or loss could be different and estimation of those values required a better understanding of the major sources and sinks of dissolved organic and inorganic
nitrogen. The isotopic composition of nitrate is useful to identify the major $NO_3^-$-N sources for the snowpacks and streams, while major sink of $NO_3^-$-N remains uncertain. Nevertheless, using the available solute chemistry and isotopic data this study presents the first quantitative estimate of the major microbial processes in the Midtre Lovénbreen catchment during the summer.


### 4.4. Microbial derived $NO_3^-$-N

Snowpack-derived $NH_4^+$-N has been suspected as the primary substrate for nitrification because it disappears very fast during snowmelt (Hodson et al., 2005b; Wynn et al., 2006; Wynn et al., 2007). This illustration is
substantiated by the discovery of ammonium oxidising archea/bacteria from supraglacial (Jakub et al., 2013) and subglacial regions (Boyd et al., 2011) respectively. However, recent studies on Midtre Lovénbreen

This is just a preview and not the published paper.





demonstrates that DON forms the largest fraction of total dissolved nitrogen in the snowpack and their loss in the subglacial flowapth and proglacial streams surpasses the $NH_4^+$-N loss (Ansari et al., 2012). Therefore, circumstantial evidence suggests that DON mineralisation and $NH_4^+$-N nitrification would have derived major
portion of microbial $NO_3^-$-N (Table 1). However, at instances gain in DON has also been noted (Figur 3a,b,c,e & 4b,c,d) inferring the possible mineralisation of particulate nitrogen, especially at the ice marginal sites (MLE3 and MLW3).

In this way, these streams carry nitrate derived from two distinct sources: 1) snowpack; 2) microbial (Figure 5). The flux of snowpack derived nitrate in the streams generally depends upon the discharge. Figure 2 shows that
the discharge of MLE and MLW decreased gradually from early to late summer. Accordingly the flux of snowpack derived nitrate in these streams also decreased. Whereas, microbially derived nitrate was independent of discharge variations and therefore remained almost constant ($1.64 \pm 1.41$ kg Day$^{-1}$ for MLE1 and $1.41 \pm 1.43$ kg Day$^{-1}$ for MLW) (Figure 2 and 6). These values of nitrification demonstrate a hitherto unknown level of microbial activity and looping effect. In which a cyclic conversion of same nitrogen between organic and
inorganic pools takes place several times over a flowpath. The downstream decrease in $\delta^{18}O(NO_3)$ and increase in $\delta^{15}N(NO_3)$ with almost constant $NO_3^-$-N provides a supporting evidence for the occurrence of looping of in-stream nitrogen pool.

*4.5. $NO_3^-$-N loss due to microbial assimilation*

Based upon the dissolved oxygen and stable isotope studies Wynn et al. (2007) first indicated the presence of denitrification activity in subglacial water. According to them some part of Midtre Lovénbreen glacier bed host a perennial melting zone maintained by high pressure. During four to five months of the Arctic winter these
zones are completely closed by ice formations, however microbiota in the stored meltwater remains active therefore consumes dissolved oxygen gradually leading to anoxia. Once dissolved oxygen content is depleted to anoxia, denitrifiers start using nitrates to oxidise organic matter for their metabolic needs. More recently denitrification was also reported in the summer proglacial streams (Ansari et al., 2012) and was attributed as a hyporheic zone (sediments and ground water in close vicinity) process (Gooseff et al., 2004; McKnight et al.,
2004). The spatial and temporal distribution of denitrification signature in surface water is controlled by rock-water interaction and subsequent exchange of solutes between surface and groundwater that mainly depends upon hydrology (Hodson et al., 2008; Tranter et al., 2005). However, according to the modelling study by Roberts et al. (2010) assimilation is the major nitrate consuming process in the Midtre Lovénbreen. Therefore, in this study the whole calculation of the microbial produced and consumed nitrate was made on the same
assumption. In this study $NO_3^-$-N in the streams were much lower than the sum of snowpack-derived and microbial derived $NO_3^-$-N. As no inorganic process is known that may amount to such efficient loss of $NO_3^-$-N ($1.39 \pm 1.41$ kg Day$^{-1}$ for MLE and $1.35 \pm 1.43$ kg Day$^{-1}$ for MLW), microbial utilisation in the flowpaths (assimilation or denitrification) appear most plausible explanation. Similar to the nitrate production, nitrate assimilation/denitrification was also remained consistent through most of the summer.

**5. Conclusion**

The solute chemistry and stable isotope investigation of the snowpack and stream samples demonstrates a large
scale microbial $NO_3^-$-N production ($1.64 \pm 1.41$ kg Day$^{-1}$ for MLE and $1.41 \pm 1.43$ kg Day$^{-1}$ for MLW) and assimilation ($1.39 \pm 1.41$ kg Day$^{-1}$ for MLE and $1.35 \pm 1.43$ kg Day$^{-1}$ for MLW) in the glacier catchment. The microbial nitrate production and consumption are several times higher than the actual nitrate flux of streams therefore infer that microbes play a much greater role in the glacial nitrogen cycling than ever considered before. Nitrate production and consumption both occurred on the side by side that probably maintained the
overall low nitrate concentration in the streams. Snowpack derived $NH_4^+$-N and DON were the major substrate for the nitrification processes. As during most of the summer nitrate production and consumption were almost consistent this reflected a fully grown and maintained microbial community. Furthermore, this study shows that nitrate production in the streams was slightly higher than the consumption, but it remains unclear how much of the assimilated nitrate converted to particulate and how much recycled back in the form of total dissolved
nitrogen ($NH_4^+$-N, $NO_3^-$-N, DON). Therefore, further studies focussed to understand the fate of assimilated nitrate is required and that will help to understand the looping of nutrients in the glacial flowpath and total nitrogen export from the catchment.


This is just a preview and not the published paper.





**Acknowledgements**

I gratefully appreciate A.J.Hodson, T.H.E. Heaton, Tristram D.L IrwvineFynn, Aga Nowak, Filip Oulehla, William Crowe, Jan Kaiser, Alina Marca-Bell, Morthekai, for their generous support in-field off-field and
laboratory. We are also thankful to the NERC British Arctic Station for providing accommodation, lab facility and logistical support. This work was supported by NSINK project (EU Marie Curie action plan, project no: R/123386).







This is just a preview and not the published paper.





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

This is just a preview and not the published paper.





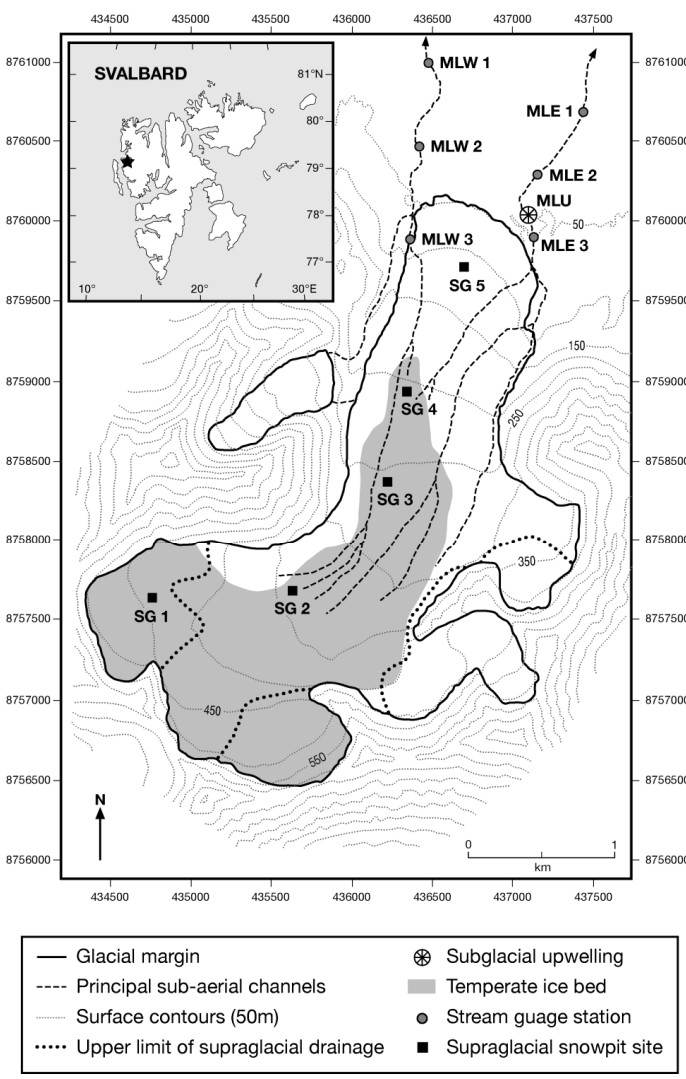


**Figure 1.** Midtre Lovénbreen map of sampling set up, UPW is the suglacial upwelling site and MLE3, MLE2, MLE1 are downstream sampling sites on the eastern stream. MLW3, MLW2, and MLW1 are downstream sampling sites on the western stream. SG1-SG5 are supraglacial snowpit sites.


This is just a preview and not the published paper.



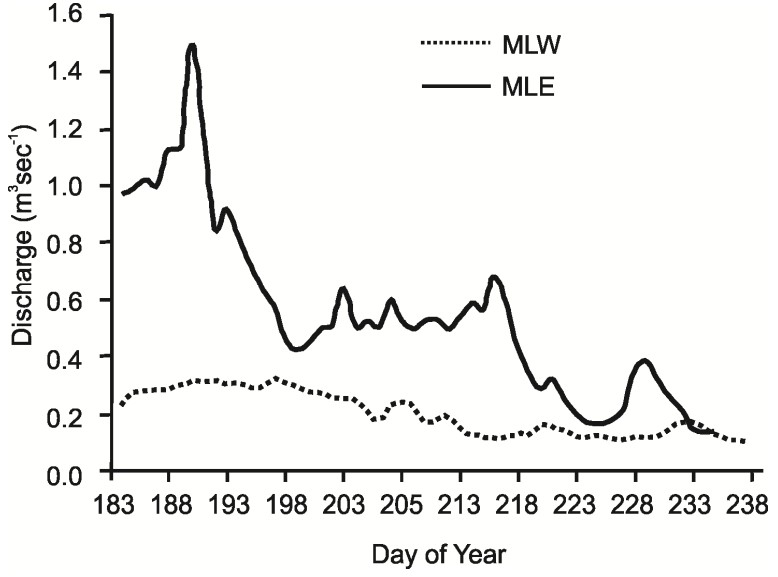

**Figure 2**. Temporal variations of average daily discharge at MLE1 and MLW1.



This is just a preview and not the published paper.





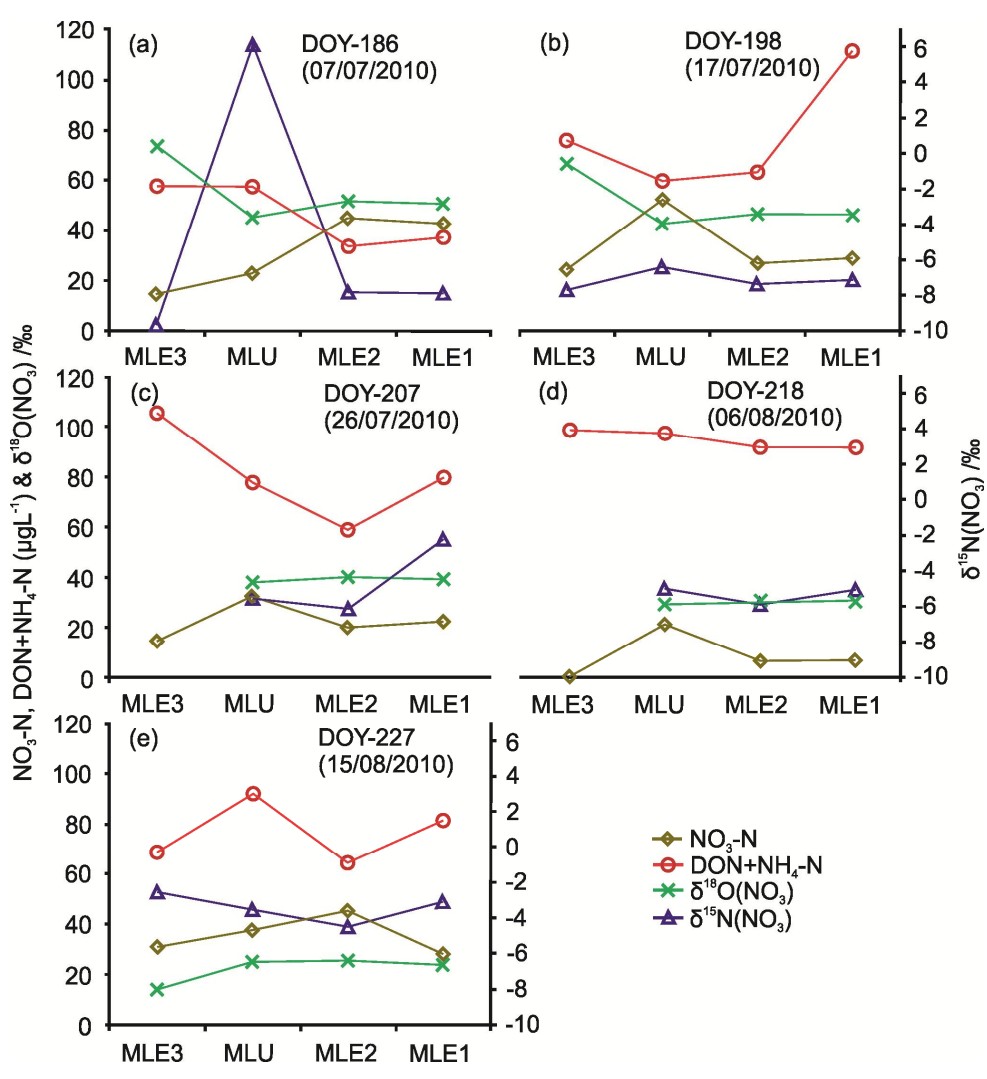

**Figure 3.** Spatial variations of $NO_3^--N$, $DON+NH_4^+-N$, $\delta^{15}N(NO_3)$, and $\delta^{18}O(NO_3)$ in MLE on (actual dates are given in bracket): (a) DOY 186, (b) DOY 198, (c) DOY 207, (d) DOY 218, and (e) DOY 227.


This is just a preview and not the published paper.



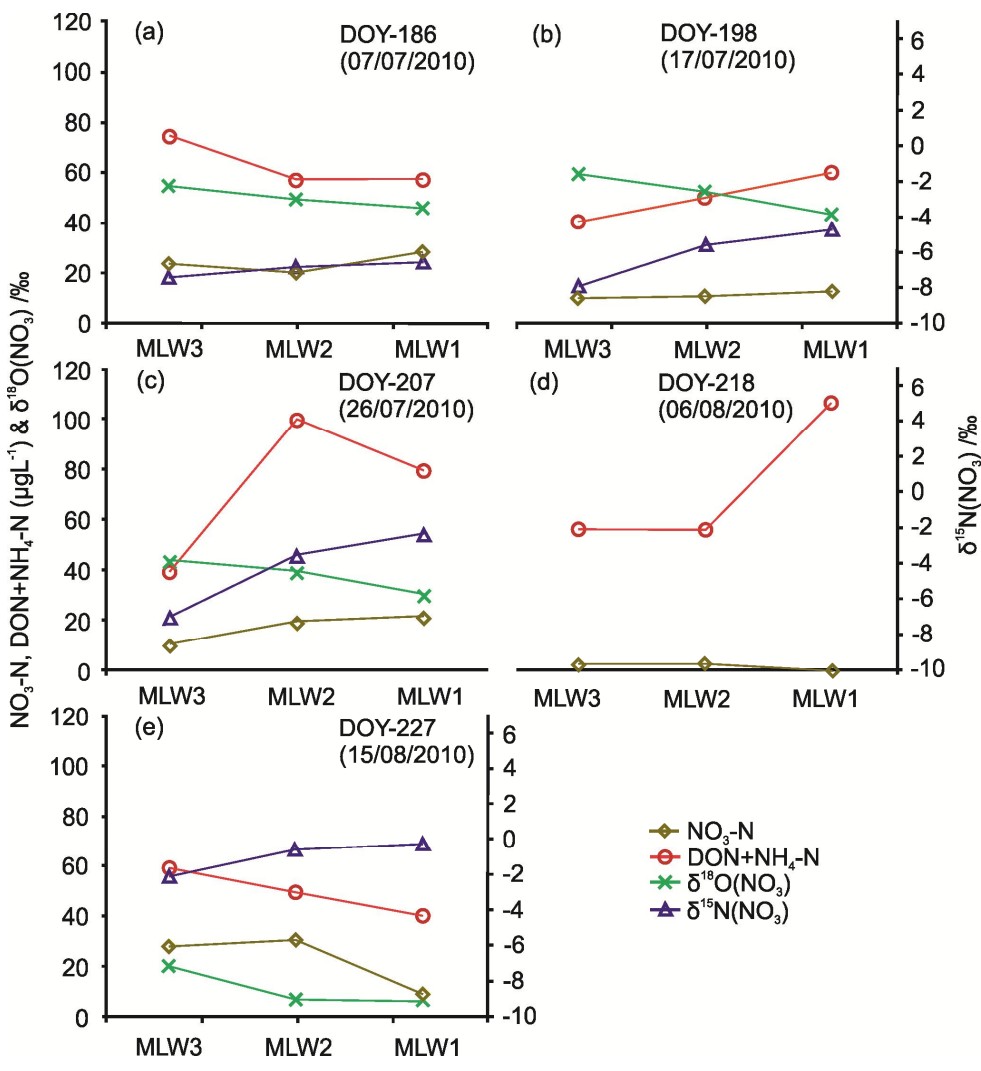

**Figure 4.** Spatial variations of $NO_3^-$-N, $DON+NH_4^+$-N, $\delta^{15}N(NO_3)$ and $\delta^{18}O(NO_3)$ in MLW on (actual dates are given in bracket): (a) DOY 186, (b) DOY 198, (c) DOY 207, (d) DOY 218, and (e) DOY 227.

This is just a preview and not the published paper.





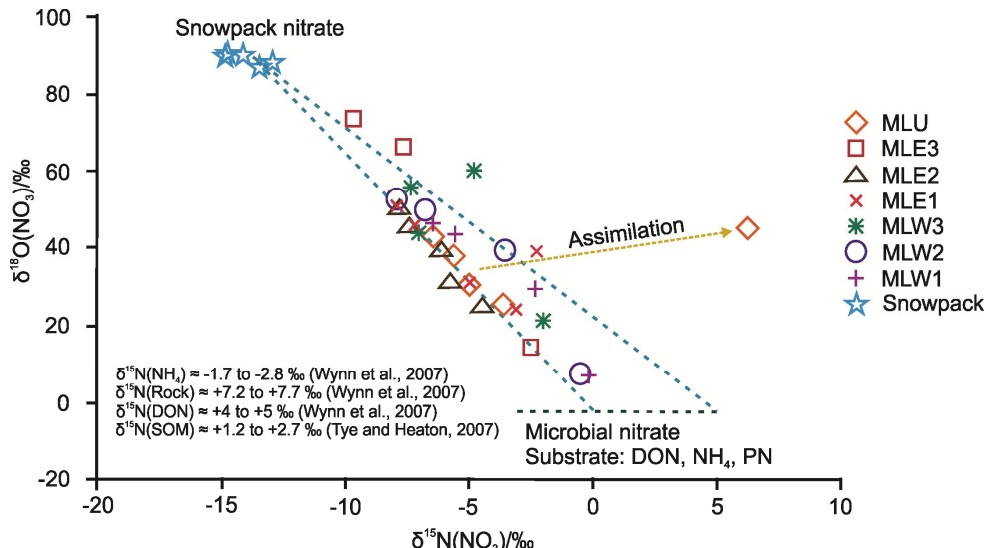

**Figure 5.** $\delta^{15}N(NO_3)$ versus $\delta^{18}O(NO_3)$ plot for the snow and stream samples.


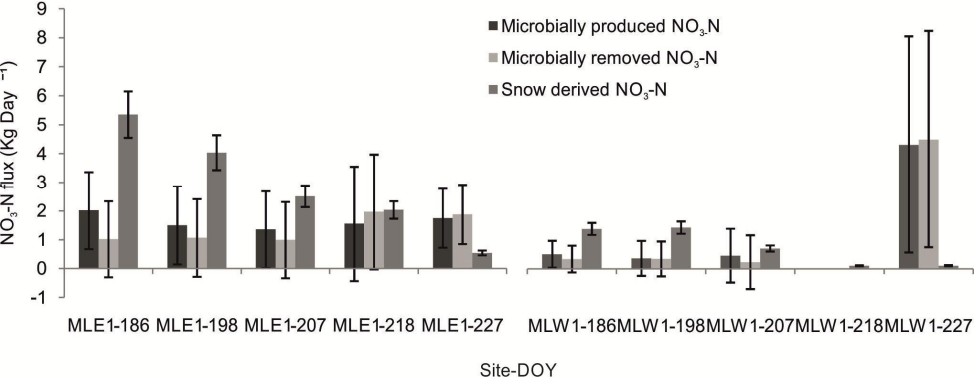

**Figure 6.** Temporal variation in the snowpack derived, microbially produced and removed nitrate in the MLE and MLW.



**Table 1**. Solute concentration and isotopic composition of the snowpacks and streams. The $S-NO_3^--N$ and $S-NH_4^+-N$ denote the snow derived $NO_3^--N$, $NH_4^+-N$ and DON respectively.

This is just a preview and not the published paper.





| Site-DOY | Actual date (dd/mm/yyyy) | Na⁺ ($mgL^{-1}$) | K⁺ ($mgL^{-1}$) | $Mg^{2+}$ ($mgL^{-1}$) | $Ca^{2+}$ ($mgL^{-1}$) | $Cl^-$ ($mgL^{-1}$) | $NO_3^-$-N ($\mu gL^{-1}$) | $NH_4^+$-N ($\mu gL^{-1}$) | DON ($\mu gL^{-1}$) | S-$NO_3^-$-N ($\mu gL^{-1}$) | S-$NH_4^+$-N ($\mu gL^{-1}$) | $\delta^{15}N(NO_3)$ /‰ | $\delta^{18}O(NO_3)$ /‰ | $\delta^{18}O(H_2O)$ /‰ |
|---|---|---|---|---|---|---|---|---|---|---|---|---|---|---|
| SG1 | | 0.93 | 0.08 | 0.43 | 1.06 | 1.47 | 20.26 | 19.44 | 80.31 | | | -13.05 | 88.08 | -10.71 |
| SG2 | | 1.08 | 0.06 | 0.47 | 0.46 | 0.86 | 12.87 | 23.55 | 93.58 | | | -14.31 | 90.19 | -10.85 |
| SG3 | | 0.68 | 0.05 | 0.36 | 0.46 | 1.10 | 20.26 | 26.11 | 73.63 | | | -13.55 | 87.28 | -10.40 |
| SG4 | | 0.69 | 0.04 | 0.38 | 0.59 | 1.22 | 30.02 | 17.79 | n.a. | | | -14.83 | 90.08 | -10.91 |
| SG5 | | 0.64 | 0.05 | 0.36 | 0.53 | 1.06 | 29.34 | 18.45 | n.a. | | | -14.95 | 89.71 | -11.61 |
| MLU-186 | 05/07/2010 | 2.34 | 1.91 | 3.87 | 8.24 | 2.68 | 22.73 | 11.61 | 45.66 | 53.37 | 51.66 | 6.24 | 45.37 | -11.7 |
| MLU-198 | 17/07/2010 | 1.57 | 1.15 | 2.73 | 14.85 | 1.85 | 52.44 | 4.45 | 43.12 | 36.90 | 35.72 | -6.42 | 42.66 | -11.3 |
| MLU-207 | 26/07/2010 | 1.29 | 1.01 | 2.53 | 3.56 | 1.33 | 32.41 | 4.12 | 73.47 | 26.56 | 25.71 | -5.51 | 37.96 | -10.7 |
| MLU-218 | 06/08/2010 | 1.34 | 0.98 | 2.64 | 18.10 | 1.46 | 21.92 | b.d. | 98.08 | 29.10 | 28.16 | -6.91 | 30.21 | -10.9 |
| MLU-227 | 15/08/2010 | 2.13 | 1.45 | 4.22 | 6.18 | 2.10 | 37.88 | 3.05 | 89.08 | 41.86 | 40.52 | -3.51 | 25.33 | -11.3 |
| MLE3-186 | 05/07/2010 | 0.33 | 0.30 | 0.41 | 7.75 | 0.51 | 12.00 | 6.84 | 48.47 | 10.13 | 9.81 | -9.62 | 73.53 | -10.5 |
| MLE3-198 | 17/07/2010 | 0.28 | 0.27 | 0.41 | 7.34 | 0.52 | 13.00 | 8.32 | 67.56 | 10.33 | 10.00 | -7.65 | 66.43 | -10.2 |
| MLE3-207 | 26/07/2010 | 0.28 | 0.38 | 0.48 | 9.45 | 0.45 | 14.61 | 7.82 | 97.56 | 8.95 | 8.66 | n.a. | n.a. | -9.5 |
| MLE3-218 | 06/08/2010 | 0.26 | 0.46 | 0.58 | 12.24 | 0.56 | 1.02 | 5.19 | 93.89 | 11.18 | 10.82 | n.a. | n.a. | -10.5 |
| MLE3-227 | 15/08/2010 | 0.90 | 1.00 | 1.87 | 6.25 | 1.51 | 31.03 | b.d. | 68.97 | 29.97 | 29.01 | -2.53 | 14.50 | -10.7 |
| MLE2-186 | 05/07/2010 | 1.25 | 1.15 | 2.22 | 12.35 | 1.51 | 45.21 | 4.61 | 20.18 | 30.11 | 29.14 | -7.81 | 51.63 | -11.4 |
| MLE2-198 | 17/07/2010 | 0.95 | 0.74 | 1.64 | 4.56 | 1.23 | 26.63 | 5.68 | 57.68 | 24.52 | 23.73 | -7.41 | 46.49 | -10.8 |
| MLE2-207 | 26/07/2010 | 0.62 | 0.56 | 1.25 | 13.73 | 0.81 | 20.06 | 5.35 | 54.58 | 16.07 | 15.55 | -6.07 | 40.31 | -10.2 |
| MLE2-218 | 06/08/2010 | 0.66 | 0.59 | 1.37 | 15.91 | 0.80 | 7.43 | 3.79 | 98.78 | 15.97 | 15.45 | -5.74 | 31.56 | -10.5 |
| MLE2-227 | 15/08/2010 | 1.82 | 1.34 | 3.78 | 8.76 | 1.85 | 45.38 | | 74.62 | 36.88 | 35.69 | -4.42 | 25.51 | -11.3 |
| MLE1-186 | 05/07/2010 | 1.31 | 1.23 | 2.43 | 2.36 | 1.57 | 42.88 | 6.42 | 30.70 | 31.33 | 30.32 | -7.85 | 50.71 | -11.1 |
| MLE1-198 | 17/07/2010 | 1.22 | 0.81 | 1.67 | 10.09 | 1.14 | 28.80 | 9.96 | 101.23 | 22.76 | 22.02 | -7.11 | 46.21 | -10.6 |
| MLE1-207 | 26/07/2010 | 0.66 | 0.61 | 1.35 | 15.94 | 0.81 | 22.52 | 7.82 | 29.66 | 16.09 | 15.57 | -2.19 | 39.31 | -9.9 |
| MLE1-218 | 06/08/2010 | 0.70 | 0.62 | 1.49 | 19.32 | 0.80 | 7.73 | 5.44 | 86.83 | 15.97 | 15.45 | -4.93 | 30.99 | -10.6 |
| MLE1-227 | 15/08/2010 | 1.93 | 1.46 | 4.42 | 14.02 | 1.83 | 28.50 | b.d. | 81.50 | 36.36 | 35.19 | -3.07 | 24.12 | -11.2 |
| MLW3-186 | 05/07/2010 | 0.93 | 0.56 | 1.29 | 10.71 | 1.03 | 24.59 | 5.85 | 69.56 | 20.48 | 19.82 | -7.32 | 55.68 | -10.7 |
| MLW3-198 | 17/07/2010 | 0.26 | 0.11 | 0.20 | 3.26 | 0.46 | 10.18 | 7.49 | 32.32 | 9.19 | 8.89 | -7.90 | 59.81 | -9.9 |
| MLW3-207 | 26/07/2010 | 0.21 | 0.13 | 0.24 | 3.73 | 0.34 | 10.00 | 4.12 | 35.49 | 6.68 | 6.47 | -7.01 | 43.20 | -9.5 |
| MLW3-218 | 06/08/2010 | 0.18 | 0.13 | 0.21 | 3.55 | 0.29 | 2.97 | 3.38 | 103.66 | 5.72 | 5.54 | n.a. | n.a. | -10.6 |
| MLW3-227 | 15/08/2010 | 0.69 | 0.32 | 0.54 | 8.29 | 1.26 | 28.77 | b.d. | 61.23 | 25.04 | 24.23 | -2.01 | 21.00 | -11.3 |
| MLW2-186 | 05/07/2010 | 0.70 | 0.62 | 1.28 | 11.59 | 0.96 | 21.41 | 6.34 | 52.25 | 19.05 | 18.44 | -6.77 | 50.09 | -10.7 |
| MLW2-198 | 17/07/2010 | 0.44 | 0.35 | 0.72 | 7.66 | 0.51 | 11.14 | 5.19 | 43.67 | 10.22 | 9.89 | -5.58 | 52.51 | -10.2 |
| MLW2-207 | 26/07/2010 | 0.79 | 0.39 | 0.86 | 8.56 | 0.72 | 18.92 | 4.28 | 96.79 | 14.26 | 13.80 | -3.56 | 39.17 | -9.8 |

This is just a preview and not the published paper.




| | | | | | | | | | | | | | | |
|---|---|---|---|---|---|---|---|---|---|---|---|---|---|---|
| MLW2-218 | 06/08/2010 | 0.39 | 0.35 | 0.74 | 7.24 | 0.39 | 2.82 | 3.71 | 53.48 | 7.69 | 7.44 | n.a. | n.a. | -10.5 |
| MLW2-227 | 15/08/2010 | 1.18 | 1.23 | 3.32 | 10.26 | 1.28 | 31.19 | b.d. | 48.81 | 25.41 | 24.59 | -0.52 | 7.58 | -10.7 |
| MLW1-186 | 05/07/2010 | 0.79 | 0.74 | 1.67 | 18.45 | 1.12 | 29.06 | 7.41 | 48.00 | 22.29 | 21.57 | -6.47 | 46.87 | -10.5 |
| MLW1-198 | 17/07/2010 | 0.49 | 0.52 | 1.16 | 10.87 | 0.61 | 12.68 | 5.85 | 73.97 | 12.14 | 11.75 | -4.70 | 43.81 | -10.2 |
| MLW1-207 | 26/07/2010 | 0.46 | 0.51 | 1.01 | 11.34 | 0.50 | 21.20 | 3.13 | 76.48 | 9.97 | 9.65 | -2.36 | 29.67 | -9.8 |
| MLW1-218 | 06/08/2010 | 0.42 | 0.49 | 1.07 | 10.13 | 0.47 | 0.39 | 5.68 | 51.35 | 9.33 | 9.03 | n.a. | n.a. | -10.5 |
| MLW1-227 | 15/08/2010 | 1.66 | 1.55 | 4.54 | 20.53 | 1.44 | 9.77 | 3.95 | 37.28 | 28.69 | 27.77 | -0.17 | 6.91 | -10.5 |

## Appendix 1

**(A)** Error propagation formula for equation (2)

$$\Delta C_{st} = \sqrt{\left(\partial C_{asm/den}\right)^2 + \left(\partial C_{mi}\right)^2 + \left(\partial C_{sn}\right)^2}$$

**(B)** Error propagation formula for equation (8)

$$\Delta C_{mi} = \left[ \left(\frac{\partial C_{mi}}{\partial \delta^{15}N_{sn}}\right)^2 \times (\Delta\delta^{18}O_{sn})^2 + \left(\frac{\partial C_{mi}}{\partial C_{sn}}\right)^2 \times (\Delta C_{sn})^2 \right.$$
$$+ \left(\frac{\partial C_{mi}}{\partial \delta^{15}N_{sn}}\right)^2 \times (\Delta\delta^{15}N_{sn})^2 + \left(\frac{\partial C_{mi}}{\partial \delta^{18}O_{mi}}\right)^2 \times (\Delta\delta^{18}O_{st})^2$$
$$+ \left(\frac{\partial C_{mi}}{\partial \delta^{15}N_{mi}}\right)^2 \times (\Delta\delta^{15}N_{mi})^2 + \left(\frac{\partial C_{mi}}{\partial \delta^{18}O_{mi}}\right)^2 \times (\Delta\delta^{18}O_{mi})^2$$
$$\left. + \left(\frac{\partial C_{mi}}{\partial \delta^{15}N_{mi}}\right)^2 \times (\Delta\delta^{15}N_{st})^2 \right]^{1/2}$$

**(C)** Error propagation formula for equation (9)

$$\Delta F = |F| \times \sqrt{\left(\frac{\partial Q}{Q}\right)^2 + \left(\frac{\partial c}{c}\right)^2}$$

