# Peer review of "Stable isotopic evidence for high microbial nitrate throughput in a High Arctic glacial catchment"

_The Cryosphere, 2016_

## Referee Comment (RC1) · Anonymous Referee #1 · 10 May 2016

General comments

This manuscript uses mass balance and isotopic analyses to examine the role of microorganisms in transforming nitrogen in proglacial streams running from a glacier in the High Arctic, Svalbard. As such, it is within the remit for Cryosphere. As seen by the similarity in titles, it is very much a follow up paper to A. Ansari's previous Biogeochemistry paper from 2013 'Stable isotopic evidence for nitrification and denitrification in a high Arctic glacial ecosystem'. The key question is whether this new paper adds sufficient new insight into glacial nutrient transformations to enable publishing in Cryosphere.

In summary, the data extends the dataset of the 2013 paper by:

1. Studying the identical suite of analyses (geochemistry , NO3- isotopic values and

discharge) of snow and proglacial streams in another year (2010 compared to 2009) 2. Studying further sample sites downstream from the first proglacial sampling points (in contrast, the 2013 paper also studied upstream supraglacial sites).

The interpretation extends the 2013 paper by:

1. Tabulating the amount of excess/depleted $NO_3^-$ in proglacial waters relative to snowpack $NO_3^-$, by normalizing against the conservative tracer $Cl^-$ (in contrast, the 2013 paper discussed this briefly in the text, finding a small xs rather than small decrease in $NO_3^-$)

2. Using a more detailed isotopic model to attempt to quantify the relative rates of microbial assimilation and nitrification (in both papers the % $NO_3^-$ produced along the transects is produced).

Overall, I do not find the findings of the new manuscript sufficiently strong or original to merit publication in Cryosphere in its present form, although the data, with additional analysis and interpretation, certainly can be published, although where will depend on the strength of new analysis and combination with previous papers (particularly the 2013 paper mentioned above). Here are my principal reservations/questions and suggestions:

Specific comments

1. The principal new conclusion that the new paper makes (based on isotopic mass balance calculations) is that there is (according to e.g. the abstract) 'fast in-stream recycling of assimilated $NO_3$-N' with 'overwhelming amounts of $NO_3$-N production and assimilation reveals a hitherto unknown level of microbial processing in the Arctic glacial ecosystem'. I have the following issues with these conclusions:

a) In the prior 2013 paper, the author (I think correctly) states that "…until our understanding of isotopic fractionation and exchange improves (a requirement for laboratory investigation that is beyond the scope of the present study), some caution is required

when interpreting Fig. 5 (analogous to part of Fig 6 in the new paper). Yet in the new paper, this caution is ignored with the calculation of exact relative rates of assimilation and nitrification, with errors based on the precision of measurements (Appendix) rather than the almost certainly far larger yet unknown errors for the different fractionation steps themselves. Without adequate knowledge of the isotopic fractionation steps in microbial glacial systems, I don't see how such precise estimates can be made. How does the author justify this change in approach, given no further insight into these issues since his previous paper?

b) Fast in stream recycling of NO3-N (presumably through NH4+ or organic N back to NO3-) will have different isotopic effects for the N and O isotopes. For O, the signature will be derived from either water or O2, and hence is not a closed system. For N, he is assuming (e.g. for complete recycling) a closed system. In this case, he will need to take into account Rayleigh fractionation in a closed system rather than the more simple mass isotopic approach he has taken.

c) Before recycling of NO3- can be assumed (as in abstract), the possibility of independent NO3- sources need to be taken into account, in particular from dissolved NH4, and DON (and particulate N, which was not measured). Rather than just focus on an excess or depletion of NO3-, a dissolved N mass balance should be carried out and included in Table 1. Is there an actual increase or decrease in total dissolved nitrogen? Can all the excess NO3- be from e.g. oxidation of DON or NH4+ rather than recycling?

d) It appears to be assumed throughout the manuscript that all microbial induced changes take place within the stream itself, by which I assume that the author means the water column. What about the pore water in underlying sediment? What about the potential for any lateral shallow groundwater transport? In line 116 you state that you 'believe that . . .discharge. . .remains unchanged from MLW3 to MLW1…..'. Without having measured comparable discharge at downstream sites, is it possible to disregard the possibility of shallow groundwater flow influencing geochemical and isotopic data?

2. There is insufficient comparison to the similar 2013 paper. This starts in the introduction, with the statement 'it has been generally considered that due to low temperature biotic impacts on nitrogen cycling in these streams have low quantitative significance' – this in direct opposition to his own 2013 paper where he states the remarkably similar conclusion that microbial processes produce up to 95% of the nitrate in the same streams. While I realise that the author is trying to differentiate between the studies, a stronger approach would be to directly combine and compare the results. The same overlapping datasets exist in 2 seasons , but apparently show some subtle seasonal differences e.g. the 2009 data shows an apparent $NO_3^-$ excess, while the 2010 new data shows slight decrease. The 2009 data shows additional data for denitrification, the 2010 does not. Why? From reading both papers again, the largest isotopic changes may occur close to the glacier (from supraglacial waters to proglacial, rather than proglacial downstream changes documented in the submitted manuscript), again why? Comparing datasets and looking at these subtle differences might potentially produce some more robust novel insight into N cycling.

3. The author uses isotopic values for snow as comparison to the proglacial streams, but presumably ice melt will have an increasing contribution as the season progresses. Were the isotopic values and concentrations of ice melt measured, and how would these affect the interpretations?

Technical comments

1. The detail given for the denitrifier method is not necessary, a similar approach should be used to the 2013 paper where basic details are given with citation.

2. The abstract needs substantial reworking. For example, it does not give the site or country where fieldwork took place.

3. I found the graphs in Figure 3+4 hard to read, with too much information. It would be clearer to e.g. separate the isotopic from compositional data and put them side by side. They should also have errors bars on, or where they are within the scale of the

symbol it should be stated. Compositional and isotopic values for snowmelt (and ice melt if available) should be plotted on the y axis, as this will clearly show the proportion of changes that occur in supraglacial to proglacial environments.

4. The 'S' columns in Table 1 should be immediately adjacent to their respective proglacial analyte to help readability. A new column should be also added for total nitrogen (sum of $NO_3^-$, $NH_4^+$ and DON) to enable a better mass balance estimate to be made.

5. Data from 2013 and new data should be combined in Figure 5 to aid seasonal and spatial interpretations.

---

## Referee Comment (RC2) · Anonymous Referee #2 · 11 May 2016

**General Comments:**  This manuscript uses an isotopic mass balance approach to quantify microbial production and consumption in a High Arctic glacial catchment.  The premise of the article is that isotopes of nitrate, along with concentration data, can be used to quantitatively estimate microbial nitrate production and consumption.  This is an excellent premise, worthy of publication in The Cryosphere.

The quantitative isotope mass balance approach is both a strength and a weakness as applied in the study.  The attempt is noteworthy, but the execution is somewhat lacking.  While the author does a nice job of propagating analytical error and standard deviations on means, error associated with other assumptions is ignored.  I'm surprised, for example, that the author does not explore the uncertainty associated with the 1:1 assumption for isotopic fractionation of $^{15}$N and $^{18}$O associated with assimilation and denitrification.  The paper the author cites for using this ratio discusses the uncertainties associated with this ratio.  In their 2012/2013 Biogeochemistry paper, the author invokes a different ratio (by a factor of 2), at least in the case of denitrification.  I found the terminology with regard to microbial 'removal' of nitrate confusing.  In some cases assimilation and denitrification are treated together, which is not unreasonable as these processes may exhibit similar isotopic fractionation effects.  In other cases, however, based on a modeling study, assimilation (also termed consumption in the manuscript) is assumed to be the dominant process.  In Figure 6, this is 'microbially removed' N.  Clear use of terminology and stating of assumptions would be beneficial.   I also don't understand why the author chose a single value for $\delta^{15}$N of microbial nitrate rather than propagating the uncertainty in this value as derived from the $\delta^{15}$N of reduced N sources.  Similarly, the variation in the oxygen isotopic composition of water could be propagated through estimation of $\delta^{18}$O of microbial nitrate.

I have read and cited the author's excellent Biogeochemistry paper on similar samples from the same study site.  I was surprised that there wasn't more reference to the previous paper.  Without such discussion I had a hard time determining if the work was a significant advance or incremental.  The greatest advance I saw was the attempt at quantification via the isotopic mass balance approach.  As such, however, all major sources of uncertainties should be propagated through the analysis.

The conclusion that microbial production is significant is not particularly novel in light of the author's earlier work.  However, the finding of how tightly cycled nitrate consumption may be with production is important, but deserving of a more robust quantitative approach that considers significant sources of uncertainty.

I believe this paper is suitable for publication in The Cryosphere, but only after significant revision.

**Specific Comments:**

Abstract:  Please state where the study took place.

Ln 19.  I don't understand the sentence starting with "These overwhelming amounts….".  Do you mean similar rates of production and consumption?

Methods: Surely we don't need this much material on the denitrifier technique.  Please cite another paper and only detail any deviations from the standard technique.

Equation 1.  Please site the conservative Cl assumption when this equation is first presented.

Lns 301 to 304.  Values given don't match Table 1.  You are supposed to be giving the ranges for two isotopes at MLU and other sites, but only two isotopic ranges are provided.  The reference to Figure 2 is incorrect.  I believe you mean Figure 5.

Ln 323.  Is it Ansari et al. 2012 or 2013?

Figure 1. You have the label MLU on the Figure (and in other plots) but UPW in the caption.

Figures 3,4.  These figures are too busy.  I'd consider separating the isotope and concentration data.

Figure 5.  I found the assumption of assimilation being more important than denitrification unconvincing.  Please label both as possibilities on the plot.  I'd prefer to see a field rather than a line for the presumed microbial nitrate source.

Table 5.  Define S-NO3 and S-NH4 in a footnote.  Put these columns next to their respective measured values and provide additional columns with the delta between the two.

---

## Referee Comment (RC3) · Anonymous Referee #3 · 31 May 2016

Dr. A. H. Ansari has used isotopic mass balance techniques to study the role of microorganisms in transforming nitrogen in High Arctic Glacial catchment. At the outset I must say that this work is based on nice premise and tries to address production and consumption of nitrate in this area. However, I also have a similar issue as pointed out by other reviewers. Does this study contribute something new compared to previous work by the author? He fails to clearly make out the conclusions and increase in our knowledge from his previous work. I have following specific comments:

1. The abstract uses sentences like "overwhelming amounts of ". This is rather subjective. Please try to be quantitative if you are claiming to provide evidence for 'high microbial nitrate throughput'.

2. I was wondering if the description about the study site and sample collection sites

could be made a bit clearer. Right now you appear to get lost at different abbreviations.

3. I do not understand why author has decided to put the method for denitrification in such great detail. This method has been established by someone else whom you have followed. Should not the author make it shorter with appropriate references? Instead they should mention where they have performed this measurement.

4. Some of the information is repeating in results and study area section.

5. Is the author proposing the mass balance equations for the first time during this study? They should again provide the proper references. In this section there are some sentences which are open ended (e.g., 323 -325). Also, He should try to make it reader friendly.

6. The discussion needs significant decluttering. It starts without the overall premise and major findings. They have divided it in different sub-headings without justifying connection to each other.

7. The major assumption by the author during the calculation is that change in microbial assimilation/denitrification is 1:1 where they have cited Kendal et al. 2007. What if this is not followed in the glacial settings author is working on. I think it would be wise to discuss about the uncertainties associated with this assumption and its possible effects on overall results and findings.

8. Author has considered the nitrogen transformation processes within the water column of stream only. What about the contribution from sediments or groundwater discharge? Is there a potential that these components may have role in modifying the isotopic composition?
* * *